# Self-medication practices to prevent or manage COVID-19: A systematic review

**Alvaro Quincho-Lopez[1], Christeam A. Benites-Ibarra[2], Maryori M. Hilario-Gomez[1], Renatta Quijano-Escate[3], Alvaro Taype-Rondan[4]***

**1** Universidad Nacional Mayor de San Marcos, Sociedad Científica de San Fernando, Lima, Peru, **2** Sociedad Científica de Estudiantes de Medicina de la Universidad Nacional del Santa, Nuevo Chimbote, Ancash, Peru, **3** Universidad Nacional San Luis Gonzaga, Sociedad Científica de Estudiantes de Medicina de Ica, Ica, Peru, **4** Universidad San Ignacio de Loyola, Unidad de Investigación Para la Generación y Síntesis de Evidencias en Salud, Lima, Peru

* alvaro.taype.r@gmail.com

**Data Availability Statement:** All relevant data are within the paper and its Supporting Information files.

## Abstract

### Background

Previous studies have assessed the prevalence and characteristics of self-medication in COVID-19. However, no systematic review has summarized their findings.

### Objective

We conducted a systematic review to assess the prevalence of self-medication to prevent or manage COVID-19.

### Methods

We used different keywords and searched studies published in PubMed, Scopus, Web of Science, Embase, two preprint repositories, Google, and Google Scholar. We included studies that reported original data and assessed self-medication to prevent or manage COVID-19. The risk of bias was assessed using the Newcastle–Ottawa Scale (NOS) modified for cross-sectional studies.

### Results

We identified eight studies, all studies were cross-sectional, and only one detailed the question used to assess self-medication. The recall period was heterogeneous across studies. Of the eight studies, seven assessed self-medication without focusing on a specific symptom: four performed in the general population (self-medication prevalence ranged between <4% to 88.3%) and three in specific populations (range: 33.9% to 51.3%). In these seven studies, the most used medications varied widely, including antibiotics, chloroquine or hydroxychloroquine, acetaminophen, vitamins or supplements, ivermectin, and ibuprofen. The last study only assessed self-medication for fever due to COVID-19. Most studies had a risk of bias in the "representativeness of the sample" and "assessment of outcome" items of the NOS.

**Funding:** The author(s) received no specific funding for this work.

**Competing interests:** The authors have declared that no competing interests exist.

## Conclusions

Studies that assessed self-medication for COVID-19 found heterogeneous results regarding self-medication prevalence and medications used. More well-designed and adequately reported studies are warranted to assess this topic.

## 1 Introduction

With the progression of the COVID-19 pandemic, several medications have been proposed as potential candidates for this disease [1], most of which resulted in little or no benefit for the patients [2, 3] or even in harms [4]. For example, hydroxychloroquine gained wide attention as a possible treatment for COVID-19 due to favorable results found in in-vitro or small uncontrolled studies [5]. However, later, randomized trials in hospitalized patients, such as the RECOVERY trial [6] and the Solidarity trial [3], failed to find any clinical benefit compared to usual care. This is similar to what happened with azithromycin [7–10], while there are still few well-designed trials that have assessed other medications such as ivermectin [11–13] or vitamins supplements [14, 15].

Nonetheless, the fear of contracting the virus, low access to health services, and massive misinformation have led some people to self-medicate. According to the World Health Organization (WHO), self-medication "involves the use of medicinal products by the consumer to treat self-recognized disorders or symptoms." [16]. This may lead to unintended consequences, such as adverse events, unnecessary expenses, delay in attending professional evaluation, masking of symptoms, and drug interactions [17–19]. Self-medication prevalence varies according to several factors, such as the methodology used to assess self-medication [20], the population characteristics [20–22], and across different countries and contexts [23–25].

Previous studies have assessed the prevalence and characteristics of COVID-19 self-medication, reporting the use of several medications, herbal products, and dietary supplements as treatment or prevention for COVID-19 [26]. However, to date, no systematic review has summarized their findings.

Thus, in this systematic review, we aimed to assess the prevalence of self-medication for the prevention or management of COVID-19. In addition, we assessed the type of medication used, reasons to practice self-medication, from where such medications were obtained, and adverse events due to its practice.

## 2 Methods

This systematic review was reported according to the Preferred Reporting Items for Systematic Reviews and Meta-Analyses (PRISMA) guidelines (Supplementary Table S3) [27]. The study protocol has been registered at PROSPERO (CRD42021236191).

### 2.1 Data sources

Search strategies were constructed a priori using different terms related to 'COVID-19' and 'self-medication' (S1 Table). On February 4th, 2021, we searched PubMed, Scopus, Web of Science, Embase, two preprint repositories (MedRxiv and SciELO Preprints), and grey literature sources (Google and Google Scholar). No language restriction was applied, but all the included articles were found written in English or Spanish. The search was restricted to include documents published in 2020 or 2021.

## 2.2 Inclusion criteria

We included original studies, either published in scientific journals or with full text available in preprint repositories, which reported original data and assessed the use of self-medication to prevent or manage COVID-19 as a primary or secondary outcome, in participants of any age and from any location.

We considered that a study assessed the use of self-medication in any of the following situations: 1) the study reported explicitly that self-medication was assessed (regardless of the definition used), or 2) the study assessed the use of medications without medical prescription.

## 2.3 Study selection

Two authors (AQL and MHG) independently selected the studies. For this, they first screened the search results by title and abstract according to the inclusion criteria using the web-based tool Rayyan (http://rayyan.qcri.org). Later, they reviewed the full text of the relevant studies to determine whether they were appropriate for study inclusion. Discrepancies were consulted with another author (ATR) and resolved by consensus.

## 2.4 Data extraction

Two authors (CBI and RQE) independently extracted the relevant data for the review using a pre-piloted Microsoft Excel spreadsheet. Again, any discrepancies were discussed with another author (ATR) and resolved with consensus.

Data on the following variables were extracted for each study: author, year of publication, country, study design, setting, population, sample technique, date of data collection, number of participants, age, sex, how was self-medication assessed, the prevalence of self-medication, reasons for practicing self-medication, who recommended such medication, from where was the medication obtained, and adverse events related to its use.

## 2.5 Risk of bias assessment

Two authors (CBI and RQE) independently evaluated the risk of bias for each included study using the Newcastle–Ottawa Scale (NOS) adapted for descriptive cross-sectional studies [28]. This scale comprises two criteria: the selection criteria (representativeness of the sample, sample size, and nonresponders) and the outcome criteria (outcome assessment and statistical tests). Any discrepancy was discussed with another author (ATR).

## 2.6 Statistical analysis

Because studies were performed in different populations and using different definitions of self-medication, we did not perform meta-analyses and decided to present the results of each study separately.

# 3 Results

## 3.1 Study selection

A total of 765 records were found in the database search. After the duplicates were eliminated, 227 records were screened, of which 46 documents were assessed in full-text. Lastly, 38 papers were excluded (which are detailed in S2 Table), and 8 papers were included, all of which were cross-sectional studies (Fig 1).

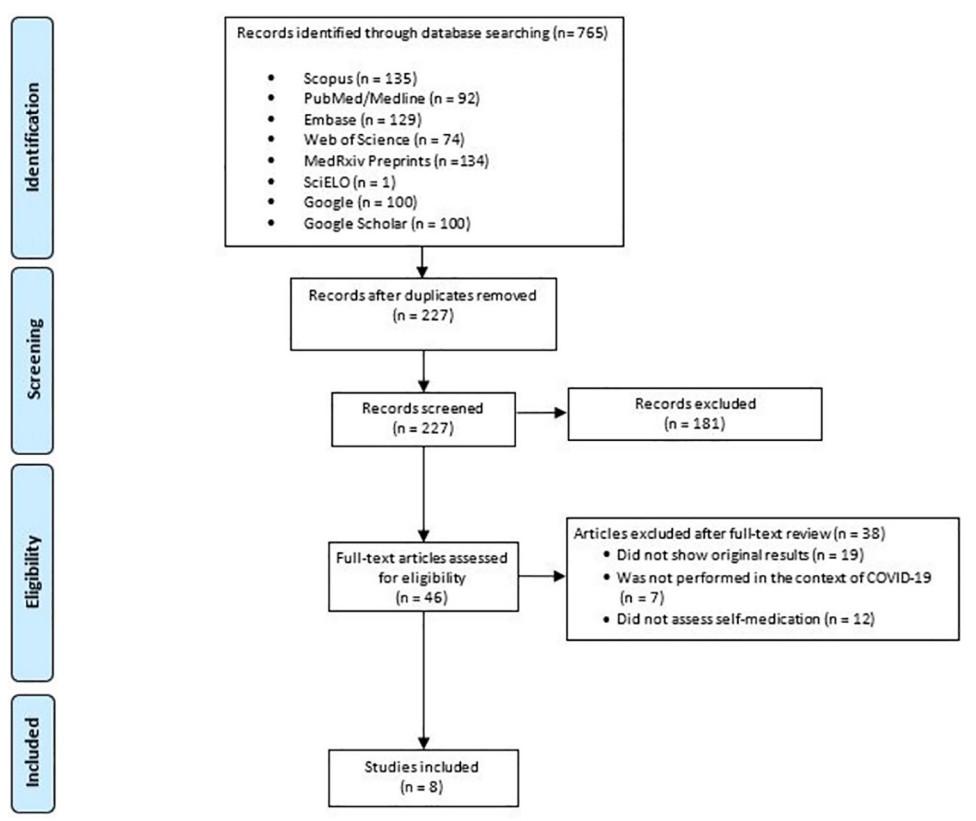

**Fig 1. Flow diagram of study selection.**

## 3.2 Methodological quality of the included studies

We assessed the risk of bias using the NOS. The study performed by Sadio [29] included two samples: one obtained using a probabilistic sample, and the other used a convenience sample. Thus, we assessed the risk of bias of these two samples separately.

Of the eight studies, only one (Sadio probabilistic sample) fulfilled the five stars in the NOS scale. In addition, the Sadio non-probabilistic sample, along with the other three studies, received four out of five stars. All studies except one (Sadio probabilistic sample) failed in fulfilling the representativeness of the sample item of the risk of bias assessment criteria (Table 1).

## 3.3 Studies characteristics

Studies characteristics are shown in Table 2. The eight included studies reported data on self-medication practices in six different countries: three from Peru [30–32], one from Bangladesh [33], one from Togo [29], one from Nigeria [34], one from Uganda [35], and one from the Kingdom of Saudi Arabia [36]. All studies were cross-sectional. Regarding their publication, five studies were published as original reports [29, 30, 33, 34, 36], one was a letter to the editor [31], and two were preprints reporting original data [32, 35]. Studies were performed from March to August 2020.

Regarding the population, five studies were conducted targeting the general population, one was conducted in hospitalized patients with COVID-19 [31], one in healthcare undergraduate students [32], and one in workers from five sectors (health care, air transport, police, road transport, and informal) [29].

**Table 1. Risk of bias assessment of included studies using the Newcastle–Ottawa Scale adapted for cross-sectional studies.**

| Study:Author (year) | Final score | Representativeness of the sample[1] | Sample size[2] | Not respondents[3] | Outcome evaluation[4] | Statistical test[5] |
|---|---|---|---|---|---|---|
| Dare (2021) | 2/5 | | | ★ | | ★ |
| Nasir (2020) | 1/5 | | | | ★ | |
| Quispe-Cañari (2021) | 4/5 | | ★ | ★ | ★ | ★ |
| Wegbom (2021) | 3/5 | | ★ | ★ | | ★ |
| Miñan-Tapia (2020) | 4/5 | | ★ | ★ | ★ | ★ |
| Sadio (2021)[6] | 5/5 | ★ | ★ | ★ | ★ | ★ |
| Sadio (2021)[7] | 4/5 | | ★ | ★ | ★ | ★ |
| Zavala-Flores (2020) | 1/5 | | | | ★ | |
| Mansuri (2020) | 4/5 | | ★ | ★ | ★ | ★ |

[1]Representativeness of the sample: One star was given to studies with random sampling or census.

[2]Sample size: One star was given to studies with justified and satisfactory sample size.

[3]Non-respondents: If comparability between respondents' and nonrespondents' characteristics was established and the response rate was satisfactory, one star was given.

[4]Outcome evaluation: If the study explicitly mentioned how self-medication was defined and how long the recall period was, one star was given.

[5]Statistical test: One star was given if it did not use a complex sample and the sample had been calculated correctly, or if it used a complex sample and such sampling was considered to estimate the self-medication prevalence.

[6]Probability sampling.

[7]Nonprobability sampling.

The eight studies included between 132 and 3,792 participants. Five studies mentioned the summary measure (mean or median) of the participants' age. Such summary measure (either mean or median) ranged from 21 to 60.3 years. All studies were performed in adults. The percentage of females ranged from 28.4% to 69.1% (Table 2).

## 3.4 Definition of self-medication

All studies assessed self-medication through self-report. The questions used to assess self-medication were explicitly mentioned only in the one study [36], while other five studies report having asked for the use of medicine without prescription [29, 31–34], and two studies gave no details regarding how self-medication was measured in their surveys [30, 35]. Moreover, three of the eight studies [30, 32, 36] reported collecting only a pre-specified list of medications.

Regarding the recall period for which self-medication was assessed: three studies assessed it during the COVID-19 pandemic [33, 35, 36], one during the COVID-19 lockdown [30], one during the past three months [32], one during the past two weeks [29], one since the COVID-19 diagnosis [31], and one did not detail the recall period [34] (Table 2).

## 3.5 Prevalence of self-medication and most frequently used medications

As showed in Table 2, We classified the eight included studies into three groups: 1) those that assessed self-medication for COVID-19 prevention or management without focusing on a specific symptom, and 2) those that assessed self-medication for a specific COVID-19 symptom.

Seven studies were included in the first group (assessed self-medication for COVID-19 prevention or management without focusing on a specific symptom). Of these, four were performed in the general population (with a range of self-medication prevalence between <4% and 88.3%) and three in specific populations (which reported a self-medication prevalence of 33.9% for hospitalized adults with COVID-19, 34.2% in adults who worked in five assessed sectors, and 51.3% in undergraduate students of health-related careers).

**Table 2. Characteristics and findings of studies that assessed self-medication to prevent or manage COVID-19.**

| Study: Author (year) | Country (city) / date of study | Subjects (age, sex) | Survey application and self-medication definition | Prevalence of self-medication for COVID-19 prevention or management (overall and most-used medications) |
|---|---|---|---|---|
| Studies that assessed self-medication without focusing on a specific symptom, performed in the general population: | | | | |
| Dare (2021)—preprint | Uganda (Western region) / from June to August 2020 | Adults from Uganda western cities (age: 43.8% between 25 and 34 years, female: 45.2%), N = 272 | • In-person and online survey (convenience sampling). | • **Overall**: 57%. |
| | | | • Self-medication assessment was not detailed. | |
| | | | • Recall period: during the COVID-19 pandemic. | |
| Nasir (2020) | Bangladesh (Dhaka city) / from April to June 2020 | Adults (age: 50.5% between 45 and 54 years, female: 55.3%), N = 626 | • Open online survey (convenience sampling). | • **Overall:** 88.3%. |
| | | | • Self-medication was defined as taking medications without prescription. | • Ivermectin (77.2%), azithromycin (54.2%), montelukast (43.1%), calcium supplements (41.4%), doxycycline (40.3%), hydroxychloroquine (20.4%). |
| | | | • Recall period: during the COVID-19 outbreak. | |
| Quispe-Cañari (2021) | Peru / June 2020 | Adults (median age: 23 years, female: 54.5%), N = 3792 | • Open online survey (convenience sampling). | • **Overall:** not reported, but taking into account the prevalence for each medication, overall prevalence was < 4%. |
| | | | • Respondents were asked to indicate whether they consumed any of the following medications: acetaminophen, ibuprofen, azithromycin, hydroxychloroquine, penicillin, or antiretrovirals. | • **For prevention:** azithromycin (0.2%), acetaminophen (0.2%), hydroxychloroquine (0.1%), antiretrovirals (0.1%), penicillin (0.1%) ibuprofen (0.03%). |
| | | | • Recall period: during the COVID-19 lockdown. | • **For management:** acetaminophen (1.7%), azithromycin (0.6%), ibuprofen (0.2%), antiretrovirals (0.1%), hydroxychloroquine (0.03%), penicillin (0.03%). |
| Wegbom (2021) | Nigeria / from June to July 2020 | Adults (mean age: 42.2 years, female: 57.1%), N = 461 | • Open online survey (convenience sampling). | • Overall: 41%. |
| | | | • Self-medication was defined as taking medications for the prevention or management of COVID-19 without prescription by medically qualified personnel. | • Vitamin C or multivitamin (21.3%), antimalarial drugs other than hydroxychloroquine and chloroquine (19.3%), amoxicillin (10.2%), ciprofloxacin (6.1%), herbal products (4.1%), metronidazole (3.5%), erythromycin (2.2%), hydroxychloroquine or chloroquine (1.3%). |
| | | | • Recall period: not detailed. | |
| Studies that assessed self-medication without focusing on a specific symptom, performed in a specific population: | | | | |
| Miñan-Tapia (2020)—preprint | Peru (Tacna) / from June to July 2020 | Undergraduate students of health-related careers from two universities (median age: 21 years, female: 69.1%), N = 718 | • Online, convenience sampling. | Overall: 51.3% |
| | | | • Self-medication was defined as using any of the pre-specified 14 medicines for COVID-19 without medical prescription. | • Acetaminophen (21.2%), ibuprofen (10.8%), dexamethasone (5.5%), aspirin (4.4%), azithromycin (2.5%). |
| | | | • Recall period: during the past three months. | |
| Sadio (2021) | Togo (Lomé) / from April to May 2020 | Adults from any of these five sectors: health care, air transport, police, road transport, and informal (median age: 36 years, female: 28.4%), N = 955 | • In-person survey through recruitment, open invitation, and random sampling (two or three stages). | • **Overall:** 34.2% |
| | | | • Self-medication was defined as the use of drugs alleged to treat or prevent COVID-19 without a physician order. | Vitamin C (27.6%), traditional medicine (10.2%), chloroquine/hydroxychloroquine (2.0%), azithromycin (1.2%). |
| | | | • Recall period: 2 weeks. | |
| Zavala-Flores (2020)–letter to the editor | Peru (Lima) / June 2020 | Hospitalized adults with COVID-19 (Mean age: 60.3 years, Female: 30.3%), N = 132 | • The authors included all patients hospitalized during a specific week. | • Overall (for COVID-19 management): 33.9% |
| | | | • Self-medication was defined as using some COVID-19 medication before hospitalization without medical prescription. | • Antibiotics (28.3%), ivermectin (20.7%), corticosteroids (17.0%), acetaminophen (12.3%), aspirin (4.7%), NSAID 84.7%), hydroxychloroquine (0.9%), others (1.9%). |
| | | | • Recall period: since the COVID-19 diagnosis. | |

*(Continued)*

**Table 2.** (Continued)

| Study: Author (year) | Country (city) / date of study | Subjects (age, sex) | Survey application and self-medication definition | Prevalence of self-medication for COVID-19 prevention or management (overall and most-used medications) |
|---|---|---|---|---|
| **Studies that assessed the use of medications for a specific symptom:** | | | | |
| Mansuri (2020) | Kingdom of Saudi Arabia / from March to April 2020 | Adults (age: 60.8% were less than 40 years, female: 60.3%), N = 388 | • An online electronic survey, convenience sampling. <br> • Self-medication was assessed using the question, "Are you self-medicating for fever?" <br> • Recall period: during the COVID-19 pandemic. | • **Overall:** 35.1% to manage fever. |

NSAID, nonsteroidal anti-inflammatory drug

Of these seven studies, six mentioned the list of medications reported by the participants. The reported medications were different across the studies: six studies reported the consumption of antibiotics, five of chloroquine or hydroxychloroquine, three of acetaminophen, three of vitamins or supplements, two of ivermectin, and two of ibuprofen.

In addition, consumption preferences varied across studies. For example, when medications were sorted in the order of highest to lowest consumption, vitamins/supplements occupied the first place in two studies, but the fourth place in another study (Table 2).

The second group (studies that assessed the use of medication for a specific symptom) included one study performed on the general population of Saudi Arabia, which found that 35.1% of the surveyed individuals used self-medication for fever [36] (Table 2).

## 3.6 Reasons to practice self-medication, from where were the medications obtained, and adverse effects

Four of the eight included studies mentioned the reasons for self-medication. However, each study assessed the reasons differently. Thus, one study referred only to the symptoms that motivated participants to self-medicate, and the other three collected a plethora of reasons, including the fear of stigmatization, fear of quarantine, affordability, the convenience of self-medication, and patients believing that the symptoms were not severe (Table 3).

Only one study [34] specified the source of the patients' self-medications. This study was performed on adults from Nigeria. The most common sources reported were the pharmacy (73.9%) and the patent medicine vendor (23.6%).

Only two studies reported whether adverse events occurred in those who self-medicate. One study performed on Undergraduate students of health-related careers found that 11.7% of those who self-medicate presented with side effects of self-medication in the past three months [32]. Nonetheless, this study did not specify the adverse effects that occurred. The second study, performed in adults from Nigeria, reported body rash (23.1%) followed by worsened condition (17.3%), yellowish eyes (7.7%), swollen face (3.8%), and vomiting of blood and severe diarrhea (5.8%) [34].

## 4 Discussion

To our knowledge, this is the first systematic review to assess the self-medication prevalence for COVID-19 prevention or management. It assessed the most important databases and

**Table 3. Reasons to practice self-medication, from where were the medications obtained, and adverse effects.**

| Study: Author (year) | Reasons to practice self-medication | Obtained from | Adverse effects |
|---|---|---|---|
| Nasir (2020) | Fever (37.6%), throat pain (28.8%), dry cough (14.2%), loss of smell (9.2%), loss of taste (3.5%), body ache (5.0%), and rarely diarrhea (1.7%) | NR | NR |
| Wegbom (2021) | Fear of stigmatization or discrimination (79.5%), fear of quarantine or self-isolation (77.3%), fear of infection or contact with a suspected or known COVID-19 patient (76.3%), delay in receiving treatment at health facilities" (55.6%), influence of friends" (55.2%), social media (54.3%), nonavailability of medications for COVID-19 treatment in the health facilities (53%), emergency illness (49.1%), delay in getting hospital services (28.1%), distance to the health facility (23%), proximity of the pharmacy (21%), nonavailability of medicine in a health facility (19.3%), and health facility charges (15.3%) | Pharmacy (73.9%), patent medicine vendor (23.6%), hospital (7.6%), hawkers (4.5%), faith-based outlets, and herbalists (2.1%, each) | Body rash (23.1%) followed by worsened condition (17.3%), yellowish eyes (7.7%), swollen face (3.8%), and vomiting of blood and severe diarrhea (5.8%). |
| Dare (2021) | Self-medication is affordable (37%) and convenient (32%), lack the means to get to the health facility/hospital (15%), fear of being diagnosed COVID-19 positive (9%), fear of visiting health facility or hospital (7%) | NR | NR |
| Miñan-Tapia (2020) | Believing that the symptoms were not severe enough to go to a doctor (64.3%), referring to family/friends that are nonmedical health professionals (34.9%), and owing to economic reasons or use of over-the-counter medications (34.9%) | NR | 11.7% participants reported adverse events (not detailed) |

NR: not reported

Note: The percentage was only based on the people who self-medicated in each study.

sources for grey literature search and preprints repositories; without language limitations. Thus, our results should reflect the state of knowledge until the search date.

Self-medication is a global phenomenon that may involve health risks at both the individual and community levels [19, 37]. Previous studies have found that self-medication is a common practice. Three systematic reviews conducted in Iran [23], Ethiopia [25], and India [24]; reported a prevalence of 53%, 44%, and 53.6%, respectively; for a recall period that ranged from a single day to 6 months.

In our review, we only found eight studies. Although the included studies were performed in six countries, the lack of studies in some regions such as North America, the Middle East, North Africa, and Oceania is evident. It is expected that the intercountry differences in drug promotion, regulations, and the possibility of accessing some medications without a prescription can influence the self-medication patterns [38, 39]. Moreover, seven of the eight included studies were performed in low- and middle-income countries. While self-medication may be higher in such countries, due to the conditions and structure of the health system [40], it is not possible to make solid comparisons due to the heterogeneity in the self-medication assessment across studies.

Some of the included studies assessed over-the-counter medications such as acetaminophen or nonsteroidal anti-inflammatory drugs, which are used for the symptom management of COVID-19, along with several other diseases. However, the included studies have also found an heterogeneous prevalence of self-medication with medications that have not proven to benefit in the prevention or management of COVID-19 (such as antibiotics, chloroquine or

hydroxychloroquine, vitamins or supplements, ivermectin, and antiasthmatics), as stated by the current COVID-19 guidelines by WHO and IDSA [41, 42].

In fact, the adverse effects of some of these medications are of great concern, such as antibiotic resistance due to extensive antibiotic use, bleeding caused by aspirin use, inhibition of the immune system caused by corticosteroids, or arrhythmia caused by hydroxychloroquine [4]. Some healthcare organizations have issued statements on self-medication. For instance, the WHO recognizes that a "successful" (i.e., beneficial) self-medication can be achieved in many countries only by improving individuals' knowledge and education level in such a way that the potential damages of this practice can be avoided [16]. Similarly, the International Pharmaceutical Federation, in tandem with the World Self-Medication Industry [43], and the World Medical Association [44], emphasizes the responsible use of non-prescription medications.

Similar to that found in previous systematic reviews [20, 21], the included studies in our review tended to use different questions to assess self-medication, or they did not specify the question used. Moreover, studies have established different recall periods, and even those that assessed self-medication since the beginning of the pandemic would have different time frames depending on the month in which they were performed. This lack of clear information prevents direct comparisons between studies and the subsequent meta-analysis of their results [45], adding a potential source of bias that affects the internal validity of the results [45, 46]. In addition, it is possible that studies that evaluated self-medication through a pre-specified list of medications reported biased prevalences, since these medications may not have included all the most commonly used medications [47].

The included studies had an important risk of bias, mainly in the "representativeness of the sample" domain of the NOS, the worst-rated item in the risk of bias assessment. This prevents the adequate extrapolation of the results [48], which should be taken into account when interpreting the results of this review.

Due to the limitations of the primary studies included in this review, more well-designed and adequately reported studies are needed, as well as the use of standardized self-medication definitions across studies. Moreover, since the COVID-19 status is changing rapidly, future studies are required to assess if there is any variation in self-medication trends between COVID-19 waves and after introducing the massive vaccination.

The findings of this review suggest that there is an important though heterogeneous prevalence of self-medication to prevent or manage COVID-19. This includes some medications that have not shown any benefit so far and may therefore expose people to unnecessary adverse events. The results vary widely across studies, suggesting that each context (region, country) has its own self-medication patterns and impact, calling on local health authorities to promote research and interventions to reduce potential self-medication adverse consequences.

## 5 Conclusion

Eight studies that assessed the use of self-medication for the prevention or management of COVID-19 were identified. The definition and recall period were heterogeneous across studies. Of the eight studies, seven assessed self-medication without focusing on a specific symptom: four performed in the general population (self-medication prevalence ranged between <4% to 88.3%) and three in specific populations (range: 33.9% to 51.3%). In these studies, the most commonly used medications varied widely, including antibiotics, chloroquine or hydroxychloroquine, acetaminophen, vitamins or supplements, ivermectin, and ibuprofen. The last study only assessed self-medication for fever due to COVID-19.

Based on four studies, fever, fear of stigmatization, affordability of self-medication, and believing that the symptoms were not severe were the most common reasons to practice self-

medication. The pharmacy was the preferred source of medications whereas body rash and worsened condition were the most frequent adverse event related to self-medication according to one and two studies, respectively. Almost all studies failed in fulfilling the representativeness of the sample item of the risk of bias assessment criteria. Thus, more well-designed and adequately reported studies are warranted in this regard.

## Supporting information

**S1 Checklist. PRISMA 2009 checklist.**
(DOC)

**S1 Table. Search strategies.**
(DOCX)

**S2 Table. List of articles excluded after full-text review.**
(DOCX)

## Author Contributions

**Conceptualization:** Alvaro Quincho-Lopez, Alvaro Taype-Rondan.

**Data curation:** Alvaro Quincho-Lopez, Christeam A. Benites-Ibarra, Maryori M. Hilario-Gomez, Renatta Quijano-Escate, Alvaro Taype-Rondan.

**Formal analysis:** Alvaro Quincho-Lopez, Christeam A. Benites-Ibarra, Maryori M. Hilario-Gomez, Renatta Quijano-Escate, Alvaro Taype-Rondan.

**Writing – original draft:** Alvaro Quincho-Lopez, Christeam A. Benites-Ibarra, Maryori M. Hilario-Gomez, Renatta Quijano-Escate.

**Writing – review & editing:** Alvaro Taype-Rondan.

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
