## [Decision Letter · Decision Letter 0]

10 Aug 2021

PONE-D-21-21439

Self-medication practices for COVID-19: a systematic review

PLOS ONE

Dear Dr Taype-Rondan,

Thank you for submitting your manuscript to PLOS ONE. After careful consideration, we feel that it has merit but does not fully meet PLOS ONE’s publication criteria as it currently stands. Therefore, we invite you to submit a revised version of the manuscript that addresses the points raised during the review process. Please submit your revised manuscript by 1st September 2021. If you will need more time than this to complete your revisions, please reply to this message or contact the journal office at plosone@plos.org. Please include the following items when submitting your revised manuscript:

We look forward to receiving your revised manuscript.

Kind regards,

Muhammad Shahzad Aslam, Ph.D.,M.Phil., Pharm-D

Academic Editor

PLOS ONE

Journal Requirements:

2. Please amend either the abstract on the online submission form (via Edit Submission) or the abstract in the manuscript so that they are identical.

Reviewers' comments:

Reviewer's Responses to Questions

**Comments to the Author**

1. Is the manuscript technically sound, and do the data support the conclusions?

Reviewer #1: Partly

Reviewer #2: Partly

Reviewer #3: Yes

Reviewer #4: Yes

2. Has the statistical analysis been performed appropriately and rigorously? 

Reviewer #1: N/A

Reviewer #2: N/A

Reviewer #3: N/A

Reviewer #4: N/A

3. Have the authors made all data underlying the findings in their manuscript fully available?

Reviewer #1: Yes

Reviewer #2: Yes

Reviewer #3: Yes

Reviewer #4: Yes

4. Is the manuscript presented in an intelligible fashion and written in standard English?

Reviewer #1: No

Reviewer #2: Yes

Reviewer #3: No

Reviewer #4: Yes

5. Review Comments to the Author

Reviewer #1: 1. Needs an extensive language editing.

2. Introduction doesn’t give enough back-ground information about the topic. It should have included a background on burden of COVID 19, self-medication, rationale/justification to conduct this review, availability of previously published review article.

3. Line 36-37, is the objective of this review just assessing the frequency of self-medication for COVID-19? Many things are mentioned in the result section. Please include all the specific objectives of this review at the end of the introduction section

4. better if ‘frequency’ is replaced by ‘prevalence’ throughout the document

5. line 50-51, in the inclusion criteria you gave the definition of self medication while you said the difference in definition is one of the limitations that hinder you from doing meta-analysis. so, remove the definition in the inclusion criteria. Ok to retain it in the introduction.

6. Include the following in the methods section: key words used for searching,

7. In your inclusion criteria nothing is said about the study participants in the retrieved studies (age (adult, children, or all), COVID status (positive, negative unknown)), region or geographic location where the studies were conducted, publication status (do you consider grey literature?)

8. ‘Drug’ and ‘medication’ are used interchangeably in the document, be consistent. I prefer ‘medication’ than ‘drug’ in your context.

9. No language restriction was applied. Do you get non-English articles? If so how or who did the translation? It should be mentioned in the methods section.

More comments are added in the attached PDF

Reviewer #2: Dear authors,

Thank you for the manuscript.

My comments are as the following:

Introduction

• Paragraph 1: The introduction part could be written with more information. For example, the authors may explain more on the examples of prevalence difference between countries, age group and occupation, in terms of the highest self-medication practices. A more thorough definition on self-medication would also be very useful, as this is the main focus of this paper, yet it is still unclear as it is only briefly mentioned in the introduction

• Paragraph 2: It would also be good if the authors described on the examples of drugs that has been proposed as the potential candidate for Covid-19, and the outcome of such treatments. The examples of direct and indirect consequences can also be further explained here, to make the introduction more informative.

• Paragraph 3: Some description on the examples of medication used as self-treatment and indication in Covid would also be informative and will make the introduction more dynamic

Method

• Data source: It would be good if the authors can include the terms used for the papers search, as this is an important component for a systematic review.

Results

Definition of self-medication

• If 9 of the 11 studies did not mention on the definition of self-medication, how did the author include these papers in the study? More explanation would be very useful. Please explain this in the manuscript. This showed the importance of providing the keywords for article search.

Who suggested the practice of self-medication ..

• In the results it stated that seeking advice from doctors and medical guideline are included as the source to the advice on self-medication. But if someone get an advice form a doctor, would it still be defined as self-medication? This goes back to the concern on the definition of the self-medication and the selection of papers in this study. Please explain.

Discussion

• It would be good if the author may emphasize on the importance of the findings and why this paper is worth to be published.

• It would be good if the author may discuss on the future study or research gap related to the topic.

Reviewer #3: This is an interesting article written in a good and readable fashion. I do have a couple of concerns, though.

Line 5 and 6: Suppose the study aimed to assess the frequency of self-medication for the prophylaxis or management of COVID-19. In that case, I was wondering why the authors included studies that assessed self-medication for all reasons?

Line 12: The authors stated that they “identified 11 studies that assessed self-medication for the prevention or management of COVID-19.” It is unclear how the authors linked the self-medication in the 7 studies that assessed “self-medication for any reason” to COVID-19. The authors’ definition of self-medication did not specify COVID-19, e.g., Quispe-Cañari (2021). The respondents may be self-medicating for another ailment.

Line 19: It is unclear why the authors chose the term frequency instead of prevalence throughout the manuscript. ‘Prevalence’ may be more appropriate to use in this context.

Line 29: Replace ‘now’ with ‘later.’

Line 33 to 35: The sentence needs to be rephrased to reveal the intended meaning. Consider rephrasing to: “Previous studies have assessed the frequency and characteristics of COVID-19 self-medication to figure out which medications are being used that are ineffective or potentially dangerous and which factors predispose people to self-medication.

Line 34: The use of the term “useless” is unsuitable for scientific writing. Consider using “ineffective.”

Line 43: I wondered if it is possible to search all these databases in a single day without any pre-defined search guide. It will be good to state that the search terms were developed apriori for better clarity to any reader.

Line 44 to 45: This is very nice. By not restricting, you avoid what is called “language bias”. However, since no language restriction was applied, it will be good for the authors to state how articles found that are not written in English were interpreted? Assuming none was found, the interpretation criteria should have been mentioned in the protocol, but unfortunately, it was not stated either in the protocol.

Line 120: The Table title does not tally with the content. It states, “…. assessed self-medication of any drug for any reason”. The title will assume that studies that assessed self-medication for the prevention and treatment of COVID-19 will not be included. The title needs to be rephrased to be all-encompassing. “Characteristics and findings of studies that assessed self-medication practices during the COVID-19 pandemic”

Table 3: In the 3rd column in the 1st row, i.e., Mansuri (2020), it is unclear how you presented the Subjects (age, sex). It seems to be in percentage. The same applies to Zavala-Flores (2020). The unit of age should be stated, i.e., years, months, e.t.c.,

Line 174: Kindly rephrase to “Only one study [22] specified the source of the patients’ self-medications.”

Line 203: Current guidelines??? It will be good to mention which guideline the authors are referring to. COVID-19 guideline?? By which body?? WHO, CDC etc.,

Throughout the manuscript: The manuscript needs detailed proofreading and revision for its English.

Reviewer #4: Thank you for the opportunity to review this article concerning the self-medication for the prophylaxis or management of COVID-19. Below are some suggestions to improve this article:

Page 3, paragraph 3: “…we aimed to assess the frequency of self-medication…”

- Please explain the reason of using the term “frequency” rather than “prevalence”.

Page 5-6, Table 1: Both the final score for Ahmed (2020) and Chauhan (2020) are zero. Please explain the rational of include these two articles in the systematic review although the scores are zero.

Page 8, paragraph 2: “Of these seven studies, only two specified which question was asked, whereas the other five studies did not detail whether participants could report self-medication only for a prespecified list of drugs or whether it was an open question.”

- The used of a prespecified list of drugs might narrow down the answers from the respondents. This subsequently might lead to biases in the study findings. This issue needs to be discussed in the discussion section.

6. PLOS authors have the option to publish the peer review history of their article (what does this mean?). If published, this will include your full peer review and any attached files.

Reviewer #1: No

Reviewer #2: No

Reviewer #3: No

Reviewer #4: No

---

## [Author Response · Author response to Decision Letter 0]

13 Sep 2021

Dear editor and reviewers,

We have carefully considered the comments and have revised the application accordingly. We believe these comments will improve our manuscript. Our responses are presented below each of the reviewer's comments.

Reviewer #1

• R1C1: Needs an extensive language editing.

o We agree with the recommendation, and a translator has checked the revised version of this manuscript. 

• R1C2: Introduction doesn't give enough back-ground information about the topic. It should have included a background on burden of COVID 19, self-medication, rationale/justification to conduct this review, availability of previously published review article.

o We agree with the recommendation; we added the following text in the second and third paragraphs of the introduction:

Line 36-40: "This may lead to unintended consequences, such as adverse events, unnecessary expenses, delay in attending professional evaluation, masking of symptoms, and drug interactions [17–19]. Self-medication prevalence varies according to several factors, such as the methodology used to assess self-medication [20], the population characteristics [20–22], and across different countries and contexts [23–25].”

Line 41-43: “Previous studies have assessed the prevalence and characteristics of COVID-19 self-medication, reporting the use of several medications, herbal products, and dietary supplements as treatment or prevention for COVID-19 [26]. However, to date, no systematic review has summarized their findings."

• R1C3: Line 36-37, is the objective of this review just assessing the frequency of self-medication for COVID-19? Many things are mentioned in the result section. Please include all the specific objectives of this review at the end of the introduction section.

o We agree with the recommendation and added these items to the objective (end of the introduction), as following:

Line 44-46: "we aimed to assess the prevalence of self-medication for the prevention or management of COVID-19. In addition, we assessed the type of medication used, reasons to practice self-medication, from where such medications were obtained, and adverse events due to its practice."

• R1C4: Better if 'frequency' is replaced by 'prevalence' throughout the document.

o We agree with the recommendation and replaced "frequency" with "prevalence" throughout the manuscript. 

• R1C5: Line 50-51, in the inclusion criteria you gave the definition of self- medication while you said the difference in definition is one of the limitations that hinder you from doing meta-analysis. so, remove the definition in the inclusion criteria. Ok to retain it in the introduction.

o In order to clarify this, we have reformulated the text at the end of the "inclusion criteria" subheading of the methods, as follows:

Line 63-65: "We considered that a study assessed the use of self-medication in any of the following situations: 1) the study reported explicitly that self-medication was assessed (regardless of the definition used), or 2) study assessed the use of medications without medical prescription."

• R1C6: Include the following in the methods section, key words used for searching.

o Thank you for your observation. The keywords used for the search in each database are available in the supplementary material (S1 Table). This is mentioned now in the second paragraph of the Methods:

Line 52-53: "Search strategies were constructed a priori using different terms related to 'COVID-19' and 'self-medication' (Supplementary Table S1)."

• R1C7: In your inclusion criteria nothing is said about the study participants in the retrieved studies (age (adult, children, or all), COVID status (positive, negative unknown)), region or geographic location where the studies were conducted, publication status (do you consider grey literature?).

o We agree with the recommendation. We have added the following in the "inclusion criteria" subsection of the Methods:

Line 61-62: "in participants of any age and from any location."

o Also, we have added the following in the "Data sources" subsection:

Line 54-55: "…and grey literature sources (Google and Google Scholar)."

• R1C8: 'Drug' and 'medication' are used interchangeably in the document, be consistent. I prefer 'medication' than 'drug' in your context.

o We agree with the recommendation and replaced 'drug' by 'medication' throughout the manuscript where appropriate.

• R1C9: No language restriction was applied. Do you get non-English articles? If so how or who did the translation? It should be mentioned in the methods section. 

o Thanks for the observation. In order to clarify this, we have added the following to the "Data sources" subsection of the Methods: 

Line 55-56: "We planned to hire a translation service if any study written in a language other than English or Spanish was found."

 Comments from the PDF file:

• R1C10: please avoid 'we' and express in passive form.

o We agree. We rephrased in passive from the first paragraph of the Results, as follows:

Line 93-95: "227 records were screened, of which 46 documents were assessed in full-text. Lastly, 38 papers were excluded (which are detailed in Supplementary Table S2), and 8 papers were included" 

• R1C11: is this mean or median.

o Thanks for the observation. To clarify this, we have improved the text of the third paragraph of the "Study characteristics" subsection of the results, as follows:

Line 139-141: "Five studies mentioned the summary measure (mean or median) of the participants' age. Such summary measure (either mean or median) ranged from 21 to 60.3 years."

• R1C12: should this study (Quispe-Cañari et al.) included in this review? b/c it doesn't assess self-medication for COVID-19.

o Thanks for the observation. However, the study of Quispe Cañari et al. assessed self-medication for COVID-19 prevention and management (table 2 of the results of the study, link: https://www.ncbi.nlm.nih.gov/pmc/articles/PMC7832015/). These are the results that we took into account.

• R1C13: is this for specific reason? no difference with studies done by Wegbom, sadio, ahmed...

o We agree. We moved Zavala-Flores et al. to the group of studies that assessed self-medication without restrictions. Also, we merged tables 2 and 3.

• R1C14: if you expect that self-medication in LMIC is higher you have to first back it up with an evidence (reference). then you can mention however...

o Thanks for the suggestion. To clarify this, in the third paragraph of the discussion now it says:

"While self-medication may be higher in such countries, due to the conditions and structure of the health system [40], it is not possible to make solid comparisons due to the heterogeneity in the self-medication assessment across studies."

40.- Torres NF, Chibi B, Middleton LE, Solomon VP, Mashamba-Thompson TP. Evidence of factors influencing self-medication with antibiotics in low and middle-income countries: a systematic scoping review. Public Health. 2019;168: 92–101. doi:10.1016/j.puhe.2018.11.018

• R1C15: inappropriate word choice (symptomatic)

o We agree. We added the following (line 210-212): "Some of the included studies assessed over-the-counter medications such as acetaminophen or nonsteroidal anti-inflammatory drugs, which are used for the symptom management of COVID-19, along with several other diseases".

Reviewer #2

Introduction

• R2C1: Paragraph 1. The introduction part could be written with more information. For example, the authors may explain more on the examples of prevalence difference between countries, age group and occupation, in terms of the highest self-medication practices. A more thorough definition on self-medication would also be very useful, as this is the main focus of this paper, yet it is still unclear as it is only briefly mentioned in the introduction.

• R2C2: Paragraph 2. It would also be good if the authors described on the examples of drugs that has been proposed as the potential candidate for Covid-19, and the outcome of such treatments. The examples of direct and indirect consequences can also be further explained here, to make the introduction more informative.

• R2C3: Paragraph 3. Some description on the examples of medication used as self-treatment and indication in Covid would also be informative and will make the introduction more dynamic.

o We agree with these 3 recommendations. Accordingly, we added the following text in the first paragraph of the introduction:

Line 25-32: "With the progression of the COVID-19 pandemic, several medications have been proposed as potential candidates for this disease [1], most of which resulted in little or no benefit for the patients [2,3] or even in harms [4]. For example, hydroxychloroquine gained wide attention as a possible treatment for COVID-19 due to favorable results found in in-vitro or small uncontrolled studies [5]. However, later, randomized trials in hospitalized patients, such as the RECOVERY trial [6] and the Solidarity trial [3], failed to find any clinical benefit compared to usual care. This is similar to what happened with azithromycin [7–10], while there are still few well-designed trials that have assessed other medications such as ivermectin [11–13] or vitamins supplements [14,15]."

o Also, we added the following to the second and third paragraphs of the introduction:

Line 36-40: "This may lead to unintended consequences, such as adverse events, unnecessary expenses, delay in attending professional evaluation, masking of symptoms, and drug interactions [17–19]. Self-medication prevalence varies according to several factors, such as the methodology used to assess self-medication [20], the population characteristics [20–22], and across different countries and contexts [23–25].”

Line 41-43: “Previous studies have assessed the prevalence and characteristics of COVID-19 self-medication, reporting the use of several medications, herbal products, and dietary supplements as treatment or prevention for COVID-19 [27]. However, to date, no systematic review has summarized their findings."

Methods

• R2C4: Data source. It would be good if the authors can include the terms used for the papers search, as this is an important component for a systematic review. 

• Thank you for your observation. The keywords used for the search in each database are available in the supplementary material (S1 Table). This is mentioned now in the second paragraph of the Methods:

o Line 52-53: "Search strategies were constructed a priori using different terms related to 'COVID-19' and 'self-medication' (Supplementary Table S1)."

Results

• R2C5: Definition of self-medication. If 9 of the 11 studies did not mention on the definition of self-medication, how did the author include these papers in the study? More explanation would be very useful. Please explain this in the manuscript. This showed the importance of providing the keywords for article search. 

o Thank you for your observation. To clarify this, we have added the following at the end of the first paragraph of the "inclusion criteria" subheading:

Line 63-65: "We considered that a study assessed the use of self-medication in any of the following situations: 1) the study reported explicitly that self-medication was assessed (regardless of the definition used), or 2) study assessed the use of medications without medical prescription."

• R2C6: Who suggested the practice of self-medication. In the results it stated that seeking advice from doctors and medical guideline are included as the source to the advice on self-medication. But if someone get an advice from a doctor, would it still be defined as self-medication? This goes back to the concern on the definition of the self-medication and the selection of papers in this study. Please explain.

o Thanks for the observation. We have discussed the subject, and we consider that the self-medication definition is incompatible with the situations in which participants consulted with physicians regarding their medications.

o Accordingly, we have added this definition in the "inclusion criteria" subsection of the methods:

o Line 63-65: "We considered that a study assessed the use of self-medication in any of the following situations: 1) the study reported explicitly that self-medication was assessed (regardless of the definition used), or 2) the study assessed the use of medications without medical prescription."

o Also, we reviewed again all the studies, and excluded 3 of them which did not match with this self-medication definition: Ahmed (2020), Kamarli Altun (2020), and Chauhan (2020). Accordingly, the review now includes only 8 studies, and all the articles have been updated.

Discussion

• R2C7: It would be good if the author may emphasize on the importance of the findings and why this paper is worth to be published.

o We agree with the recommendation. Accordingly, we added the following as the last paragraph of the discussion (right before the conclusion):

o Line 244-249: "The findings of this review suggest that there is an important though heterogeneous prevalence of self-medication to prevent or manage COVID-19. This includes some medications that have not shown any benefit so far and may therefore expose people to unnecessary adverse events. The results vary widely across studies, suggesting that each context (region, country) has its own self-medication patterns and impact, calling on local health authorities to promote research and interventions to reduce potential self-medication adverse consequences."

• R2C8: It would be good if the author may discuss on the future study or research gap related to the topic.

o Thanks for the suggestion. We added the following as the penultimate paragraph of the discussion:

o Line 239-243: "Due to the limitations of the primary studies included in this review, more well-designed and adequately reported studies are needed, as well as the use of standardized self-medication definitions across studies. Moreover, since the COVID-19 status is changing rapidly, future studies are required to assess if there is any variation in self-medication trends between COVID-19 waves and after introducing the massive vaccination."

Reviewer #3

• R3C1: Line 5 and 6 "Suppose the study aimed to assess the frequency of self-medication for the prophylaxis or management of COVID-19. In that case, I was wondering why the authors included studies that assessed self-medication for all reasons?"

o We understand the concern of the reviewer. To clarify our inclusion criteria, we have changed the redaction as follows:

Line 59-62: “We included original studies, either published in scientific journals or with full text available in preprint repositories, which reported original data and assessed the use of self-medication to prevent or manage COVID-19 as a primary or secondary outcome, in participants of any age and from any location.”

Line 63-65: “We considered that a study assessed the use of self-medication in any of the following situations: 1) the study reported explicitly that self-medication was assessed (regardless of the definition used), or 2) the study assessed the use of medications without medical prescription."

• R3C2: Line 12 "The authors stated that they "identified 11 studies that assessed self-medication for the prevention or management of COVID-19." It is unclear how the authors linked the self-medication in the 7 studies that assessed "self-medication for any reason" to COVID-19. The authors' definition of self-medication did not specify COVID-19, e.g., Quispe-Cañari (2021). The respondents may be self-medicating for another ailment".

o Thank you for your observation. To clarify this, we have added the following at the end of the first paragraph of the "inclusion criteria" subheading:

Line 63-65: "We considered that a study assessed the use of self-medication in any of the following situations: 1) the study reported explicitly that self-medication was assessed (regardless of the definition used), or 2) study assessed the use of medications without medical prescription. "

o Regarding the study of Quispe Cañari et al, it assessed self-medication for COVID-19 prevention and management (table 2 of the results of the study, link: https://www.ncbi.nlm.nih.gov/pmc/articles/PMC7832015/). These are the results that we took into account.

• R3C3: Line 19 "It is unclear why the authors chose the term frequency instead of prevalence throughout the manuscript. 'Prevalence' may be more appropriate to use in this context".

o We agree and have made such changes throughout the manuscript.

• R3C4: Line 29 "Replace 'now' with 'later'”

o As we modified the writing of the manuscript, this suggestion is not valid anymore.

• R3C5: Line 33 to 35 “The sentence needs to be rephrased to reveal the intended meaning. Consider rephrasing to: “Previous studies have assessed the frequency and characteristics of COVID-19 self-medication to figure out which medications are being used that are ineffective or potentially dangerous and which factors predispose people to self-medication”

o Thanks for the observation. We have changed the following: 

Line 2-3: “Previous studies have assessed the prevalence and characteristics of self-medication in COVID-19. However, no systematic review has summarized their findings.”

Line 41-43: “Previous studies have assessed the prevalence and characteristics of COVID-19 self-medication, reporting the use of several medications, herbal products, and dietary supplements as treatment or prevention for COVID-19 [26]. However, to date, no systematic review has summarized their findings.”

• R3C6: Line 34 “The use of the term ‘useless’ is unsuitable for scientific writing. Consider using ‘ineffective’”

o As we modified the writing of the manuscript, this suggestion is not valid anymore.

• R3C7: Line 43 “I wondered if it is possible to search all these databases in a single day without any pre-defined search guide. It will be good to state that the search terms were developed a priori for better clarity to any reader”.

o We agree with the recommendation, So, in order to guide the readers, we added the following line in “Data sources” section:

Line 52-53: “Search strategies were constructed a priori using different terms related to ‘COVID-19’ and ‘self-medication’ (Supplementary Table S1).”

• R3C8: Line 44 to 45 “This is very nice. By not restricting, you avoid what is called “language bias”. However, since no language restriction was applied, it will be good for the authors to state how articles found that are not written in English were interpreted? Assuming none was found, the interpretation criteria should have been mentioned in the protocol, but unfortunately, it was not stated either in the protocol”.

o Thanks for the observation. We added the following lines in the “Data sources” section to clarify this: 

Line 55-56: “We planned to hire a translation service if any study written in a language other than English or Spanish was found.”

• R3C9: Line 120 “The Table title does not tally with the content. It states, “…. assessed self-medication of any drug for any reason”. The title will assume that studies that assessed self-medication for the prevention and treatment of COVID-19 will not be included. The title needs to be rephrased to be all-encompassing. “Characteristics and findings of studies that assessed self-medication practices during the COVID-19 pandemic””

o We merged tables 2 y 3, and this new table 2 have the following title:

Line 131-132: “Characteristics and findings of studies that assessed self-medication to prevent or manage COVID-19.”

• R3C10: Table 3 “In the 3rd column in the 1st row, i.e., Mansuri (2020), it is unclear how you presented the Subjects (age, sex). It seems to be in percentage. The same applies to Zavala-Flores (2020). The unit of age should be stated, i.e., years, months, e.t.c.”

o We agree and have made the change, in the first row of the third column of table 3 now it says “age: 60.8% were less than 40 years, female: 60.3%”. Then, in the last row of the third column of table 2, now it says, “Mean age: 60.3% years”.

• R3C11: Line 174 “Kindly rephrase to “Only one study [22] specified the source of the patients’ self-medications””

o We agree and have made such change in the results. Please, see the second paragraph (line 180) of the “Reasons to practice self-medication, from where were the medications obtained, and adverse effects” subheading in the results.

• R3C12: Line 203 “Current guidelines??? It will be good to mention which guideline the authors are referring to. COVID-19 guideline?? By which body?? WHO, CDC etc”

o We consider the recommendation, so we rephrased the sentence (4th paragraph of the discussion). Now it says: 

Line 215-216: “As stated by the current COVID-19 guidelines by WHO and IDSA [41,42].”

• R3C13: Throughout the manuscript “The manuscript needs detailed proofreading and revision for its English”

o We agree with the recommendation, and a translator has checked the revised version of this manuscript. 

Reviewer #4

• R4C1: Page 3, paragraph 3: “…we aimed to assess the frequency of self-medication…”. Please explain the reason of using the term “frequency” rather than “prevalence”.

o We have reviewed the differences between the two terms and agree with your suggestion. So that, we replaced ‘frequency’ by ‘prevalence’ throughout the manuscript.

• R4C2: Page 5-6, Table 1: Both the final score for Ahmed (2020) and Chauhan (2020) are zero. Please explain the rational of include these two articles in the systematic review although the scores are zero.

o Due to the commentaries of another reviewer, we have realized that 3 of the initially included studies did not fulfil the self-medication definition because they included medication taken due to a prescription. We have excluded these 3 studies from the results: Ahmed (2020), Kamarli Altun (2020), and Chauhan (2020).

• R4C3: Page 8, paragraph 2: “Of these seven studies, only two specified which question was asked, whereas the other five studies did not detail whether participants could report self-medication only for a prespecified list of drugs or whether it was an open question”. The used of a prespecified list of drugs might narrow down the answers from the respondents. This subsequently might lead to biases in the study findings. This issue needs to be discussed in the discussion section.

o We agree with your suggestion and decide to add the following sentence (6th paragraph of the discussion): 

Line 232-234: “In addition, it is possible that studies that evaluated self-medication through a pre-specified list of medications reported biased prevalences, since these medications may not have included all the most commonly used medications [47].”

47.- Choi B, Granero R, Pak A. Catalog of bias in health questionnaires. Rev Costarr Salud Pública. 2010;19: 106–118. Available: http://www.scielo.sa.cr/scielo.php?script=sci_arttext&pid=S1409-14292010000200008&lng=en.

---

## [Decision Letter · Decision Letter 1]

29 Sep 2021

PONE-D-21-21439R1Self-medication practices to prevent or manage COVID-19: a systematic reviewPLOS ONE

Dear,

Thank you for submitting your manuscript to PLOS ONE. After careful consideration, we feel that it has merit but does not fully meet PLOS ONE’s publication criteria as it currently stands. Therefore, we invite you to submit a revised version of the manuscript that addresses the points raised during the review process.

 Please submit your revised manuscript by 1st November 2021. If you will need more time than this to complete your revisions, please reply to this message or contact the journal office at plosone@plos.org. Please include the following items when submitting your revised manuscript:A rebuttal letter that responds to each point raised by the academic editor and reviewer(s). You should upload this letter as a separate file labeled 'Response to Reviewers'.A marked-up copy of your manuscript that highlights changes made to the original version. You should upload this as a separate file labeled 'Revised Manuscript with Track Changes'.An unmarked version of your revised paper without tracked changes. You should upload this as a separate file labeled 'Manuscript'.

We look forward to receiving your revised manuscript.

Kind regards,

Muhammad Shahzad Aslam, Ph.D.,M.Phil., Pharm-D

Academic Editor

PLOS ONE

Journal Requirements:

Reviewers' comments:

Reviewer's Responses to Questions

**Comments to the Author**

1. If the authors have adequately addressed your comments raised in a previous round of review and you feel that this manuscript is now acceptable for publication, you may indicate that here to bypass the “Comments to the Author” section, enter your conflict of interest statement in the “Confidential to Editor” section, and submit your "Accept" recommendation.

Reviewer #1: All comments have been addressed

Reviewer #2: All comments have been addressed

Reviewer #3: All comments have been addressed

Reviewer #4: All comments have been addressed

2. Is the manuscript technically sound, and do the data support the conclusions?

Reviewer #1: Partly

Reviewer #2: Yes

Reviewer #3: Yes

Reviewer #4: (No Response)

3. Has the statistical analysis been performed appropriately and rigorously? 

Reviewer #1: N/A

Reviewer #2: Yes

Reviewer #3: N/A

Reviewer #4: (No Response)

4. Have the authors made all data underlying the findings in their manuscript fully available?

Reviewer #1: Yes

Reviewer #2: Yes

Reviewer #3: Yes

Reviewer #4: (No Response)

5. Is the manuscript presented in an intelligible fashion and written in standard English?

Reviewer #1: No

Reviewer #2: Yes

Reviewer #3: No

Reviewer #4: (No Response)

6. Review Comments to the Author

Reviewer #1: still some language editing is needed. The conclusion should contain the main findings expressed in general terms and should address all the objectives.

Reviewer #2: Dear authors

All my comments have been sufficiently addressed.

Improvement of the manuscript can be seen.

Reviewer #3: (No Response)

Reviewer #4: (No Response)

7. PLOS authors have the option to publish the peer review history of their article (what does this mean?). If published, this will include your full peer review and any attached files.

Reviewer #1: No

Reviewer #2: **Yes: **Siti Maisharah Sheikh Ghadzi

Reviewer #3: No

Reviewer #4: No

---

## [Author Response · Author response to Decision Letter 1]

1 Oct 2021

Comments from reviewer #1:

R1C1: Line 55. I think you mean ''We had planned to hire a translation service if any study written in a language other than English have been found”. This is a sentence that should have been included in the protocol. but for the manuscript, better if replaced with 'but all the included articles were found written in English'.

o We agree with the recommendation and have changed that sentence according to your suggestion. Please, see line 55-56: “No language restriction was applied, but all the included articles were found written in English or Spanish”.

R1C2: Line 76: Add ‘data on’ before this sentence. Replace ‘assessed’ with ‘extracted’.

o We agree with the recommendation. Please, see line 76: “Data on the following variables were extracted for each study: author, year of publication, country…”.

R1C3: Line 106-107: Replace ‘representing’ with ‘fulfilling the representativeness of’. Replace ‘selection’ with ‘risk of bias assessment’.

o We agree with both recommendations. Please, see lines 106-107: “All studies except one (Sadio probabilistic sample) failed in fulfilling the representativeness of the sample item of the risk of bias assessment criteria (Table 1)”.

R1C4: Conclusions: All the objectives and main findings are not addressed in the conclusion.e.g. over all range of prevalence, the type of medication used, reasons to practice self-medication...

o We agree. We rephrased and added new sentences regarding the prevalence, type of medication, reasons, source, adverse events related to self-medication and quality of the included studies. Please, see lines 251-265: “Eight studies that assessed the use of self-medication for the prevention or management of COVID-19 were identified. The definition and recall period were heterogeneous across studies. Of the eight studies, seven assessed self-medication without focusing on a specific symptom: four performed in the general population (self-medication prevalence ranged between <4% to 88.3%) and three in specific populations (range: 33.9% to 51.3%). In these studies, the most commonly used medications varied widely, including antibiotics, chloroquine or hydroxychloroquine, acetaminophen, vitamins or supplements, ivermectin, and ibuprofen. The last study only assessed self-medication for fever due to COVID-19. 

Based on four studies, fever, fear of stigmatization, affordability of self-medication, and believing that the symptoms were not severe were the most common reasons to practice self-medication. The pharmacy was the preferred source of medications whereas body rash and worsened condition were the most frequent adverse event related to self-medication according to one and two studies, respectively. Almost all studies failed in fulfilling the representativeness of the sample item of the risk of bias assessment criteria. Thus, more well-designed and adequately reported studies are warranted in this regard”.

R1C5: Line 255: Add ‘commonly’ after ‘most’.

o We agree with the recommendation. Please, see line 255: “In these studies, the most commonly used medications varied widely...”.

---

## [Decision Letter · Decision Letter 2]

18 Oct 2021

Self-medication practices to prevent or manage COVID-19: a systematic review

PONE-D-21-21439R2

Dear,

We’re pleased to inform you that your manuscript has been judged scientifically suitable for publication and will be formally accepted for publication once it meets all outstanding technical requirements.

Kind regards,

Muhammad Shahzad Aslam, Ph.D.,M.Phil., Pharm-D

Academic Editor

PLOS ONE

Additional Editor Comments (optional):

Reviewers' comments:

Reviewer's Responses to Questions

**Comments to the Author**

1. If the authors have adequately addressed your comments raised in a previous round of review and you feel that this manuscript is now acceptable for publication, you may indicate that here to bypass the “Comments to the Author” section, enter your conflict of interest statement in the “Confidential to Editor” section, and submit your "Accept" recommendation.

Reviewer #1: All comments have been addressed

Reviewer #2: All comments have been addressed

Reviewer #4: All comments have been addressed

2. Is the manuscript technically sound, and do the data support the conclusions?

Reviewer #1: Yes

Reviewer #2: Yes

Reviewer #4: Yes

3. Has the statistical analysis been performed appropriately and rigorously? 

Reviewer #1: Yes

Reviewer #2: Yes

Reviewer #4: N/A

4. Have the authors made all data underlying the findings in their manuscript fully available?

Reviewer #1: Yes

Reviewer #2: Yes

Reviewer #4: Yes

5. Is the manuscript presented in an intelligible fashion and written in standard English?

Reviewer #1: Yes

Reviewer #2: Yes

Reviewer #4: Yes

6. Review Comments to the Author

Reviewer #1: the authors have addressed all my previous comments. better if conclusions are precisely summariezed in a single paragraph.

Reviewer #2: Dear authors

All my comments have been sufficiently addressed.

Improvement of the manuscript can be seen.

Reviewer #4: (No Response)

7. PLOS authors have the option to publish the peer review history of their article (what does this mean?). If published, this will include your full peer review and any attached files.

Reviewer #1: No

Reviewer #2: No

Reviewer #4: No

---

## [Editor Report · Acceptance letter]

21 Oct 2021

PONE-D-21-21439R2 

Self-medication practices to prevent or manage COVID-19: a systematic review 

Dear Dr. Taype-Rondan:

I'm pleased to inform you that your manuscript has been deemed suitable for publication in PLOS ONE. Congratulations! Your manuscript is now with our production department. 

Kind regards, 

on behalf of

Dr. Muhammad Shahzad Aslam 

Academic Editor

PLOS ONE